# Physical, Hydric, Thermal and Mechanical Properties of Earth Renders Amended with Dolomitic Lime

**DOI:** 10.3390/ma15114014

**Published:** 2022-06-06

**Authors:** Halidou Bamogo, Moussa Ouedraogo, Issiaka Sanou, Jean-Emmanuel Aubert, Younoussa Millogo

**Affiliations:** 1Laboratoire de Chimie et Energies Renouvelables (LaCER), UFR/Sciences Exactes et Appliquées, Département de Chimie, Université Nazi BONI (UNB), Bobo-Dioulasso 01 BP 1091, Burkina Faso; 2Laboratoire Matériaux et Durabilité de Constructions (LMDC), INSA-Génie Civil, Université de Toulouse, 135 Avenue de Rangueil, CEDEX 4, F-31 077 Toulouse, France; aubert@insa-toulouse.fr

**Keywords:** earth renders, dolomitic lime, calcium silicate hydrate, magnesium silicates hydrates, physical, hydric and mechanical properties, thermal comfort

## Abstract

The global objective of this work was to manufacture resistant and durable (water resistant) earth renders with good thermal insulation. For this purpose, a medium plastic clayey soil from Kôdéni (Burkina Faso), constituted by kaolinite (62 wt.%), quartz (31 wt.%), and goethite (2 wt.%), was mixed with dolomitic lime (up to 6 wt.%) to manufacture earth renders. The mineralogical, microstructural, and chemical characteristics of dolomitic lime, as well as the physical (linear shrinkage, apparent density, and accessible porosity), hydric (water absorption test by capillarity and spray test), thermal (thermal conductivity), and mechanical (abrasion resistance, flexural, and compressive strengths) properties of the earth renders were evaluated. From these studies, it appears that the addition of dolomitic lime induces the formation of calcium silicate (CSH) and magnesium silicate (MSH) hydrates. These CSH and MSH are mainly formed from the pozzolanic reaction between finely ground quartz and the weak silica of kaolinite in basic media. These formed hydrates improve the physical, hydric, thermal, and mechanical properties of earth renders. This improvement is due to the fact that the formed CSH and MSH stick to the isolated particles of the soil, making them more compact. In view of the good mechanical strength and water resistance, as well as the low thermal conductivity, the elaborated earth renders are suitable for habitats with dry climates, such as the Sahel.

## 1. Introduction

In Burkina Faso, as in most developing countries, traditional habitats are still built using vernacular construction materials, such as the raw clayey materials used as earth renders. The use of raw clayey materials to elaborate earth renders for protecting habitats in these developing countries has some advantages, such as their good thermal comfort; however, there are important disadvantages linked to their low mechanical properties and weak water resistance. 

In order to improve upon the above properties, vegetable fibers, organic binders, and small quantities of mineral binders (cement or lime) are commonly introduced (Bamogo et al. [1]; Ashour et al. [2]; Lima et al. [3]; Palumbo et al. [4]).

In fact, Bamogo et al. [1] focused on the use of cow dung as a stabilizer in earth renders. Cow dung is an original stabilizer for earth renderings in the rural world. This work revealed that the addition of cow dung in earth renders improves the physical, hydric, thermal, and mechanical properties. Ashour et al. [2] showed that the thermal conductivity of some natural plaster materials elaborated with earth or sand mixed with three types of straw, such as as wheat straw, barley straw, and wood chips, decreased with the increase of the fiber content. Lima et al. [3] showed that the addition of natural fibers, such as goat straw and typha wool, to plasters improves their mechanical properties, and the habitats using these plasters present a thermal comfort. The physical characteristics have shown that the addition of these fibers contributes to a decrease of the linear drying shrinkage and thermal conductivity and promotes the adhesion strength of the plaster to the substrate. The investigation by Palumba et al. [4] on the incorporation of plant materials, such as barley straw and corn pith (1 and 2 wt.%), into compressed earth blocks and earth renders, focused on the study of thermal conductivity and diffusivity, as well as water vapor permeability and the moisture buffering of reinforced earth blocks. These earth materials adsorb 15.5 g more moisture than unreinforced materials. These results showed that the plant materials have a significant impact on the thermal properties and apparent density of the mixtures, but they have a limited effect on the hydric properties. This work has shown the positive impacts of these additives on these technological properties.

Nevertheless, these works have not dealt with the influence of the chemical and mineralogical compositions of the mineral and organic binders on the renders matrix. It is observed that the chemical and mineralogical compositions of these binders have an effect on the clay matrix, and this effect undoubtedly contributes to the improvement of the earth renders properties.

To our knowledge, few scientific works have been devoted to the stabilization of earth renders by dolomitic lime. The formulation of renderings based on a mixture of clay and dolomitic lime is an ancestral practice for developing countries, in general, and Burkina Faso, in particular. This ancestral practice shows that these binders have a real potential to improve the physical and mechanical properties of earth renders, as well as to protect the renders against water erosion, while ensuring good thermal comfort in the houses.

The mineralogy, chemistry, and adhesion of the binders could be the basis for the protection of these plasters against water erosion, hence the need for an in-depth scientific study on the mechanism of action of these binders on the clay matrix.

The global objective of this work was to manufacture resistant and durable (water resistant) earth renders with good thermal insulation. 

This scientific work deals with the influence of dolomitic lime (up 6 wt.%) on the physical and mechanical properties of earth renders manufactured with clayey soil from Kôdéni (western Burkina Faso). The effect of dolomitic lime additions on the erosion resistance, abrasion resistance, and thermal conductivity of earth renders was highlighted, and particular attention was paid to the formation mechanism of calcium silicate (CSH) and magnesium silicate (MSH) hydrates during earth renders manufacturing.

## 2. Raw Materials, Methods, and Experimental Procedures

### 2.1. Raw Materials

Bamogo et al. [1] previously studied the raw clayey material used for the elaboration of earth renders. This clay soil, referenced as KOD, was collected in the locality of Kôdéni (11°10′ N and 04°17′ W) in western Burkina Faso. Local populations exploit this site, essentially for the production of construction materials (earth bricks and earth renders). The mineralogical composition of the raw material has been published in a previous paper [1]. The mineralogical composition was obtained by coupling chemical, X-ray Diffraction (XRD), DSC-TGA, and Infra-Red (IR) spectra analyses. These results are presented in Table 1. The geotechnical tests on raw clayey material (granulometry, Atterberg limits, and methylene blue test) allowed us to obtain its category (Table 1) [1].

In view of obtained results and according to the German standard [5], KOD clay is suitable for the elaboration of earth renders, because it contains a high proportion of kaolinite, which will serve as a binder for the isolated particles [6], and does not contain a swelling clay species, such as montmorillonites and smectites.

The lime used in this study is commercial dolomitic lime, and it was supplied by the Coveni company in Bobo Dioulasso (Burkina Faso). 

### 2.2. Methods and Experimental Procedures

#### 2.2.1. Chemical, Mineralogical, and Microstructural Characterization of Raw Materials and Earth Renders

Chemical analysis of the dolomitic lime was carried out by inductively coupled plasma-atomic emission spectrometry (ICP-AES). A sample of raw material was crushed to particles size less than 80 µm and melted with lithium tetraborate (Li_2_B_4_O_7_) to form a glass bead. The glass bead was dissolved in a nitric acid solution, and the solution obtained was analysed by an ICP-AES device.

X-ray diffraction patterns of the dolomitic lime and renders sample powders were obtained with a Siemens D5000 type diffractometer, equipped with a monochromator using cobalt Kα radiation (Co Kα, λ = 1.789 Å) and graphite back monochromator. The range of analysis was between 5 and 75°, with a step size of 0.04°. The acquisition time of the analyses was 2 s. The analyses were all performed on powders previously crushed to particles size less than 80 µm and placed in a rotating sample holder. 

Analysis by differential scanning calorimetry (DSC) consisted of monitoring the difference in the heat flow between the two containers—one contained the analyzed sample, and the second contained the thermally inert sample (calcined alumina). The heating rate during the analyses was 10°C/min. Thermogravimetric analysis (TGA) consists of recording the variation in mass during a thermal cycle. These changes are related to chemical reactions or the departure of volatile constituents adsorbed or combined in a material. The two techniques are often applied simultaneously on the same apparatus and sample. The DSC-ATG thermograms were recorded with a Netzsch SATA 449F3 Jupiter apparatus, under an argon atmosphere between 30 and 1100 °C. A quantity between 10 and 30 mg of the sample, with a maximum particle size not exceeding 80 µm, was tested.

Infra-red spectrometry on crushed dolomitic lime (particles size less than 80 µm) was carried out on an attenuated total reflectance-Fourier transform infra-red spectrometer (ATR-FTIR Frontier, 4000–500 cm^−1^, diamond crystal-Perkin Elmer). The device used for the analysis was a Perkin Elmer L125000P Frontier.

Scanning electron microscopy (SEM) was used to explore the dolomitic lime sample surface, in order to provide information regarding the mineral’s morphology. SEM observation of this sample was carried out with JEOL 6380 LV microscope. Energy dispersive X-ray spectroscopy (EDS) was used to access to the chemical composition of the sample’s explored surface. This analysis was carried out using Brüker X Flash 6/30 detector. 

#### 2.2.2. Manufacture of Earth Rendering Mortars

The earth rendering mortar’s elaboration was carried out according to the protocol described by Bamogo et al. [1].

For the manufacturing of the raw clay renders, the KOD raw material was oven-dried at 105 °C for 24 h and crushed to particles with sizes <5 mm. The obtained clay powder was mixed with different amounts of dolomitic lime (0, 2, 4, and 6 wt.% of the dry clay). The optimal amount of water (equal to 24 wt.%) was added to the previous mixture, in order to obtain a paste that was suitable for the elaboration of earth rendering mortars. Pastes were homogenized for 15 min and stored in an hermetically sealed plastic container in a temperature-controlled room (25 °C) for 72 h to allow for ripening. Homogeneous pastes were introduced into the molds (200 × 200 × 20 mm^3^) in two layers, with series of manual shocks of 20 strokes total per layer. The molds were kept in the shade (to avoid thermal shock) for 72 h. After demolding, the obtained coatings were kept for 90 days at room temperature (30 ± 7 °C, with an average humidity of 32 ± 10%), in order to avoid the appearance of cracks. To perform the various tests, three samples of each formulation were developed. This required the formulation of 24 samples of different earth renders formulations. The proportions of the mixtures used for the manufacture of earth rendering mortars are given in the Table 2.

#### 2.2.3. Physical, Hydric, Thermal, and Mechanical Characterization of Earth Renders

Linear shrinkage, apparent density, and capillary water absorption tests of prismatic samples (4 × 4 × 16 cm^3^) of earth renders were carried out according to the German standard DIN 18947 [5], EN 1015-10 [7], and EN 15801 CEN 2009 [8], respectively.

Accessible porosity, pH measurement, thermal conductivity, spray test, abrasion resistance, three-point flexural, and compressive strengths were performed according to the methodologies described by Bamogo et al. [1].

## 3. Results and Discussion

### 3.1. Dolomitic Lime Chemical, Mineralogical, and Microstructural Characteristics

The chemical, mineralogical, and microstructural characterization of dolomitic lime was performed after carbonation, because dolomitic lime is partially hydrated and carbonated, since portlandite, calcite, and magnesium carbonate were observed on the diffractogram.

Chemical analysis of the dolomitic lime shows that it is composed mainly of 54.83 wt.% of calcium oxide (CaO), 37.25 wt.% of magnesium oxide (MgO), and 7.92 wt.% of silica (SiO_2_), and it contains insoluble residues formed essentially of quartz.

X-ray diffraction pattern of dolomitic lime (Figure 1) shows the presence of calcium oxide (CaO), portlandite (Ca (OH)_2_), calcium carbonate or calcite (CaCO_3_), magnesium carbonate (MgCO_3_), and quartz (Q).

The IR spectrum of dolomitic lime is shown in Figure 2. The spectrum shows the presence of prominent vibration bands of calcite: 1414 cm^−1^ (Ca-O vibration) [9] and portlandite: 874 cm^−1^ (Ca-OH elongation vibration) [10] and 3687 cm^−1^ (OH elongation vibration) [10,11]. The band around 957 cm^−1^ corresponds to the vibration bands of the Mg-O bonds of magnesium carbonate [12].

The DSC-TGA curves for dolomitic lime are given in Figure 3. They show an endothermic peak around 120 °C, associated with a mass loss of 1.9 wt.%, attributable to the departure of moisture water. The endothermic peak around 384 °C corresponds to the decarbonation of magnesium carbonate [12]. This phenomenon is associated with a loss of mass of 9.7 wt.%. The endothermic peak around 440 °C is due to the dehydroxylation of the portlandite and hydroxide magnesium [6]. This phenomenon is followed by a loss of mass of 3.7 wt.%. The chemical reactions associated to these phenomena are: Ca(OH)_2_(s) → CaO(s) + H_2_O(g) (1)
Mg(OH)_2_(s) → MgO(s) + H_2_O(g) (2)
MgCO_3_(s) → MgO(s) + CO_2_(g)(3)

The endothermic peak around 734 °C, associated with an important mass loss of 10.8 wt.%, is related to the decarbonation of calcite [6], following the chemical Equation (4):CaCO_3_(s) → CaO(s) + CO_2_(g)(4)

Scanning electron microscopy (SEM) of dolomitic lime (Figure 4a) shows a rather irregular morphology, in the form of aggregates, with a remarkable presence of rhombohedral platelets, which is a characteristic of calcite. In addition, the presence of small grains (white color) was observed, which is characteristic of magnesia (magnesium oxide) [12]. The analysis by energy dispersive spectrometry EDS (Figure 4b) that was carried out on this dolomitic lime shows that it contains not only a high amount of calcium oxide (CaO) and magnesium oxide (MgO) but also a significant proportion of silica (SiO_2_).

### 3.2. Mineralogy of Earth Renders Amended with Dolomitic Lime

X-ray diffraction and differential scanning calorimetry (DSC) are the two techniques used to explore the mineralogical characteristics of the amended earth renders with dolomitic lime. The different tests were performed on the powders (ϕ < 80 μm) of each grade after 90 days of curing.

X-ray diffraction was performed on the sample powders of the earth renders reinforced with dolomitic lime, in order to identify new formed crystalline phases. The diffractograms of these samples are presented in Figure 5. A fine examination of the diffractograms shows the presence of kaolinite, quartz, and goethite, as well as calcite and portlandite. The intensity of the main peaks of portlandite and calcite increased with dolomitic lime additions, while those for kaolinite and quartz decreased slightly with up to 4 wt.% dolomitic lime addition. Diffractograms of samples containing 4 and 6 wt.% dolomitic lime showed diffused peaks with very low intensity for 48° < 2θ < 50°. These peaks could be attributed to weakly crystallized calcium silicate hydrates (CSH) [6]

The DSC curves of the earth renders samples (Figure 6) highlight the endothermic phenomena. The endothermic peak around 80 °C corresponds to the hygroscopic water loss phenomena of the clay soil. The large endothermic peak between 150–300 °C corresponds to the dehydration of calcium silicate (CSH) and magnesium silicate (MSH) hydrates [13,14,15]. The peak at 502 °C is attributable to the dehydroxylation of portlandite (CH) and kaolinite, as well as its transformation into metakaolinite [6,16,17]. The weak peak around 578 °C characterizes the allotropic transformation of quartz α to quartz β. The endothermic phenomenon around 678 °C corresponds to the decomposition of calcite to produce quicklime (CaO) [6,18].

To elucidate the formation of CSH and MSH, the evolution of the intensities of the basal peak of kaolinite (2θ = 12.36°) and main peak of quartz (2θ = 26.6°), as a function of dolomitic lime amount (Figure 7), were highlighted. The observation of the above peaks intensities reveals a decrease with dolomitic lime additions up to 4 wt.%. Beyond this value, an increase of these peaks is observed. The decrease of these peaks would be due to the consumption of silica by the silicates of the kaolinite and fine crushed quartz of raw clayey materials. This pozzolanic-type reaction takes place due to the presence of the portlandite Ca(OH)_2_ and Mg(OH)_2_ obtained by the hydration of the dolomitic lime. The basic character of such a medium favors this pozzolanic reaction, as shown by the pH increase of the clay–dolomitic lime mixtures (Figure 8). Additionally, the grinding of the samples favors this reaction due to the quartz fine grain surface activation, which makes them more reactive. The increase in peaks intensities with the important amount of dolomitic lime could be explained by the excess silica (7.92 wt.%) contained in the latter. In this case, all the formed portlandite and hydroxide magnesium were consumed during the pozzolanic reaction. It was also observed that the intensity of the main quartz and basal kaolinite peaks decreased with up to 4 wt.% dolomitic lime addition. The partial solubilization of silica from kaolinite is facilitated by its disordered structure [19]. It can be deduced that the formation of CSH and MSH is mainly due to the pozzolanic reaction involving the fine ground quartz. The reactions involved in the formation of CSH and MSH can be summarized as follows:-dissociation of portlandite and hydroxide magnesium from dolomitic lime in aqueous medium, according to chemical Equations (5) and (6), respectively:
Ca(OH)_2_ ⇆ Ca^2+^ + 2OH^−^
(5)
Mg(OH)_2_ ⇆ Mg^2+^ + 2OH^−^
(6)-Solubilization in a basic medium of fine ground quartz or silica from silicates of the kaolinite, according to chemical Equation (7):
SiO_2_ + 2OH^−^ ⟶ H_2_SiO_4_^2−^(7)-Reaction in aqueous medium of the formed calcium ions Ca2+and Mg2+ with solubilized silica (H2SiO42−) to form CSH and MSH, according to chemical Equations (8) and (9):
xCa^2+^ + H_2_SiO_4_^2−^ + 2(x-1) OH^−^ + yH_2_O ⟶ CSH (8)
x Mg^2+^ + H_2_SiO_4_^2−^ + 2(x-1) OH^−^ + yH_2_O ⟶ MSH (9)

### 3.3. Physical, Hydric, Thermal, and Mechanical Properties of Earth Renders Amended with Dolomitic Lime

#### 3.3.1. Physical Properties of Earth Renders

The linear shrinkage, apparent density, and accessible porosity of earth renders have been evaluated. The obtained results of these physical properties are reported in Table 3.

The linear shrinkage of the earth renders samples (Table 3) decreases with dolomitic lime additions. This decrease is due to the fact that, with the addition of dolomitic lime, there is the formation of CSH, which glue soil-isolated particles, thus limiting the shrinkage phenomenon. The results obtained in this study are similar to those reported by Attoh-Okine et al. [20] on the stabilization of lateritic soil by lime. The linear shrinkage values of dolomitic lime-amended earth renders were lower than the maximum value required for earth renders, which is 3%, according to the German standard [5].

The results in Table 3 show an increase in accessible porosity and decrease in apparent density with dolomitic lime addition. This increase in accessible porosity is justified by the formation of flakes in the earth renders. This is due to the attraction of calcium and magnesium ions, which are produced during the dissociation of dolomitic lime by the surface of negatively charged kaolinite grains, creating small pores [21]. The decrease in the apparent density is due to the increase in accessible porosity (Table 3). The accessible porosity and apparent density evolve inversely. The same observations were reported by Bamogo et al. [1] on earth renders reinforced with cow dung.

#### 3.3.2. Hydric and Thermal Properties of Earth Renders

The hydric behavior of earth renders is a very important criterion for their acceptability in the rehabilitation of buildings. For this purpose, the capillary water absorption and spray tests have been carried out on the manufactured earth renders.

The evolution of the water absorption coefficient by the capillarity of earth renders with dolomitic lime additions is presented in Figure 9. The water absorption coefficient of the earth renders amended with dolomitic lime decreases with up to 4 wt.% of dolomitic lime. This decrease is due to the formation of CSH- and MSH-binding soil-isolated particles increasing the cohesive force of the particles in this clayey matrix, considerably reducing the kinetics of the capillary rise of water in the composite materials [6,15]. Beyond 4 wt.% of dolomitic lime, the water absorption coefficient increases slightly. This slight increase is explained by the excessive formation of portlandite and calcite in the clayey matrix [6]. The results obtained in this study are somewhat different from those reported by some authors on the stabilization of earth renders by hydraulic and air lime [22,23,24,25,26]. However, in this study, soil rich in sand was used, and this contributed to the production of more CSH and MSH, which helped to reduce the porosity of the sample by binding the isolated particles [15]. On the other hand, the difference in results is related to the intrinsic properties of the lime used for the manufacturing of earth renders [27].

Under the effect of rains, earth renders samples can undergo more or less pronounced disintegration, depending on their quality. The impact of rains on different earth renders was evaluated by the spray test, using artificial rain. The evolution of the mass loss of the earth renders samples, as a function of the dolomitic lime content after the erosion test, is presented in Figure 10. This figure shows that the raw earth renders were much more eroded than those reinforced with dolomitic lime. It also shows that the mass loss of the earth renders samples decreased with the increase of dolomitic lime, up to an optimal value of 4 wt.%. This decrease is due to the formation of calcium silicate (CSH) and magnesium silicate (MSH) hydrates, which glue the isolated particles by contributing to the reduction of size of open pores [6,21]. Above 4 wt.% of dolomitic lime, there is an increase in mass loss, due to the excessive formation of calcite and portlandite, making earth renders permeable to water.

Burkina Faso is a developing country and, in particular, a Sahel country, subject to high temperatures. It is confronted with the major problem of significant savings, due to energy consumption related to the use of air conditioning and ventilation systems. For this purpose, thermal conductivity was carried out with earth renders samples, in order to evaluate their thermal conductivity. The aim of the development of earth renders will be to offer a good thermal comfort to the buildings, while decreasing the expenses in electrical energies, which are expensive for the majority of the Burkina Faso population. 

The results of the thermal conductivity (λ) of the earth renders samples with dolomitic lime additions are presented in Figure 11. Generally, the thermal conductivity (λ) of the earth renders decreases when dolomitic lime content increases. This decrease in thermal conductivity could be due to the increase of closed pores filled with air, an insulating medium. The thermal conductivity of the earth renders of this study was similar to that reported by Bamogo et al. [1] and Santos et al. [26], where all formulations had values less than 0.9 W/(m.K).

#### 3.3.3. Mechanical Properties of Earth Renders

Building materials are subjected to stresses and react to simple abrasion, compression, and bending stresses. The primary quality of a building material is to resist to these effects, without being altered. The mechanical resistance is, therefore, an important parameter for the selection of building materials. Figure 12 shows the evolution of the abrasion coefficient of earth renders, as a function of dolomitic lime content. The abrasion coefficient of earth renders decreases with dolomitic lime additions up 4 wt.%. This decrease is justified by the good cohesion between soil particles and dolomitic lime, due to the formation of CSH and MSH [6,15,21]. This good cohesion generates clay–lime and/or clay–sand–lime bonds, making the grains of the earth renders more consolidated and better able to face mechanical stresses. Above 4 wt.% of dolomitic lime, the abrasion coefficient of the earth increases slightly. This slight increase in the abrasion coefficient is due to the weak cohesion between the particles, with the presence of external pores in places making the earth renders less resistant. The abrasion coefficients of dolomitic lime-reinforced renders allow them to be classified in the SI category (abrasion category with a mass loss less than 1.5 g/cm^2^), according to the German standard [5]. The results of this study were compared with those of earth renders reinforced with cow dung [1], which showed better abrasion coefficients than the earth renders amended with dolomitic lime. This is due to the good adhesion of the fibers contained in the cow dung with the clay matrix because of their rough surface and good distribution in this matrix.

The flexural and compressive strengths of earth renders with dolomitic lime additions are presented in Figure 13. The flexural and compressive strengths of the earth renders increased up to an optimal value (4 wt.%). This increase is justified mainly by a homogeneous microstructure, with a good interconnection of soil-isolated particles, due to the formation of CSH and MSH [6,15,21]. Beyond this value, the resistance of the earth renders decreases. This decrease is due, on one hand, to the excessive formation of calcite and portlandite in the clayey matrix and, on the other hand, to the increase of accessible porosity. By comparing the results obtained in this study, regarding the flexural and compressive strengths of the earth renders, with those of the literature [3,5,19], it could be concluded that earth renders reinforced with dolomitic lime meet the requirements of the mechanical strengths (flexural and compressive) defined for earth renders in the construction of habitats.

## 4. Conclusions

The main objective of this work was to manufacture resistant, durable (water resistant), and thermally-insulating earth renders. For this purpose, the clayey raw material KOD and dolomitic lime were characterized, in order to use them for the elaboration of earth renders. The results of the mineralogical and chemical characterization showed that the clay KOD was composed mainly of kaolinite (62 wt.%), quartz (31 wt.%), and goethite (2 wt.%), and the dolomitic lime was composed mainly of calcite, portlandite, and magnesium carbonate. The addition of dolomitic lime to the clayey soil induced the formation of calcium silicate (CSH) and magnesium silicate (MSH) hydrates. These CSH and MSH were mainly formed from the pozzolanic reaction between finely ground quartz and weak silica from kaolinite in basic medium. These formed hydrates improved the physical, hydric, thermal, and mechanical properties of the earth renders. This improvement was due to the formation of CSH and MSH, which play the role of binders, allowing for the cohesion of the isolated particles of the clayey matrix, making the composite materials more compact. The earth renders are characterized by good mechanical strength and water resistance, as well as low thermal conductivity; therefore, they are suitable for habitats in dried climates, such as Burkina Faso. 

The hygrothermal properties, initiated in this study by the measurement of thermal conductivity, will have to be deepened, in particular, by taking the water vapour exchanges with these earth renders into account. Tests such as the measurement of water vapour sorption–desorption and moisture buffer value (MBV) would complete the hygrothermal characterization and make it possible to show the interest of using these earth renders reinforced with dolomitic lime to improve the comfort of habitats.

## Figures and Tables

**Figure 1 materials-15-04014-f001:**
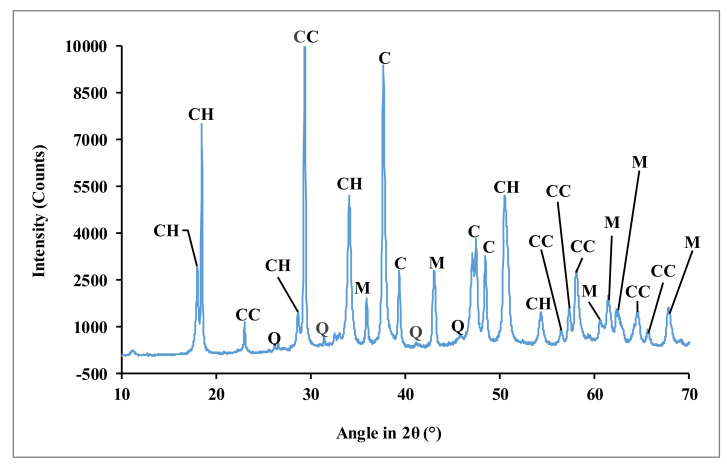
X-ray diffraction pattern of the dolomitic lime (CH: portlandite; C: calcium oxide; CC: calcite; M: magnesium carbonate; and Q: quartz).

**Figure 2 materials-15-04014-f002:**
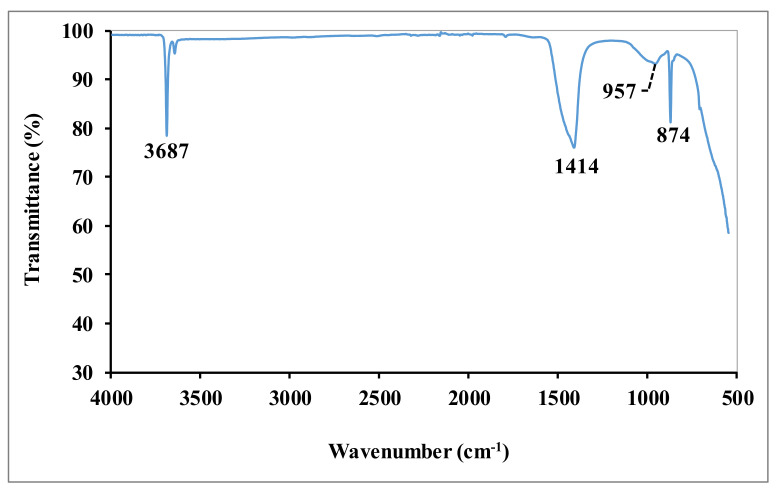
FTIR spectrum of the dolomitic lime.

**Figure 3 materials-15-04014-f003:**
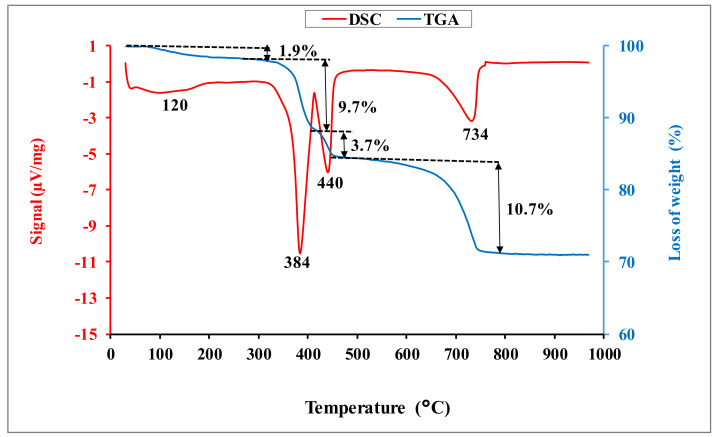
DSC-TGA curves of the dolomitic lime.

**Figure 4 materials-15-04014-f004:**
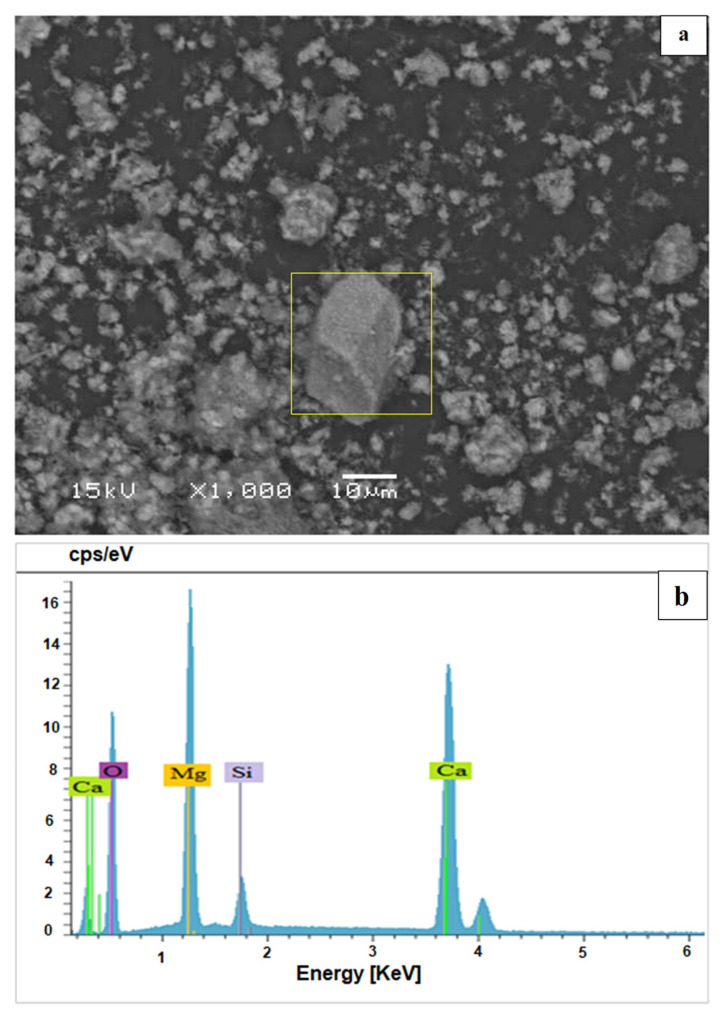
SEM image (**a**) and EDS spectrum (**b**) of dolomitic lime.

**Figure 5 materials-15-04014-f005:**
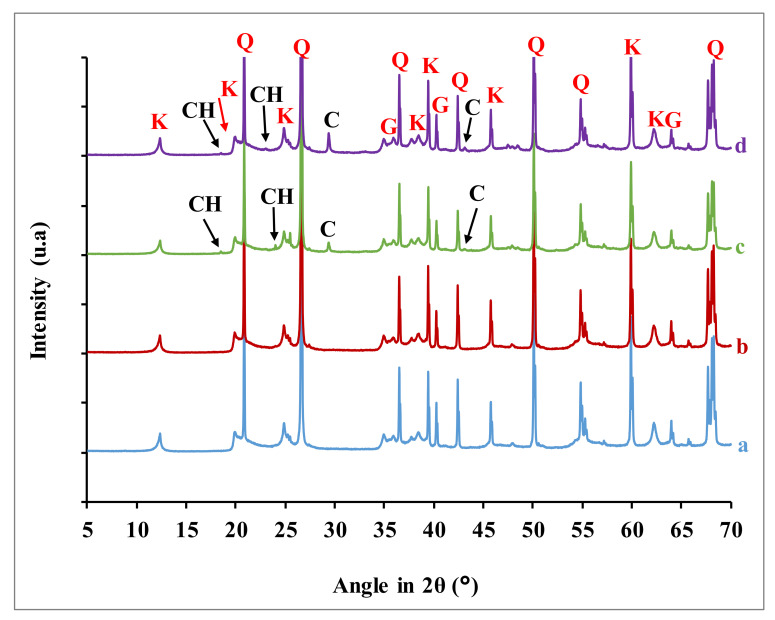
X-ray diffraction patterns of the earth renders samples. a) Raw earth renders; earth renders with b) 2 wt.% dolomitic lime, c) 4 wt.% dolomitic lime, and d) 6 wt.% dolomitic lime (K: kaolinite, Q: quartz, G: goethite, CH: portlandite, and C: calcite).

**Figure 6 materials-15-04014-f006:**
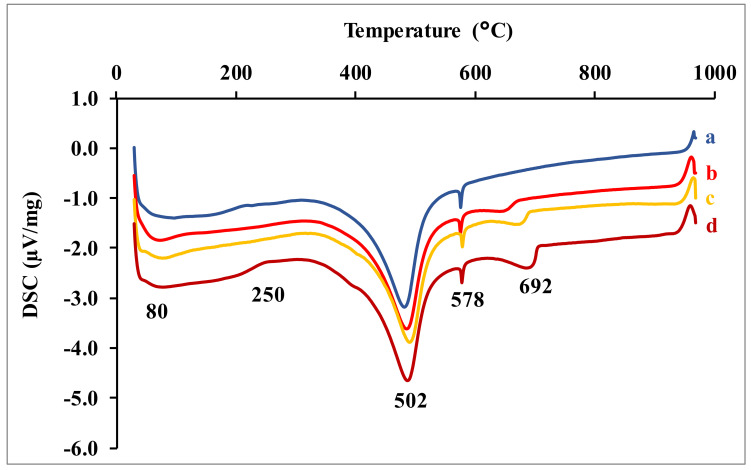
DSC curves of earth renders samples. a) Raw earth renders; earth renders with b) 2 wt.% dolomitic lime, c) 4 wt.% dolomitic lime, and d) 6 wt.% dolomitic lime.

**Figure 7 materials-15-04014-f007:**
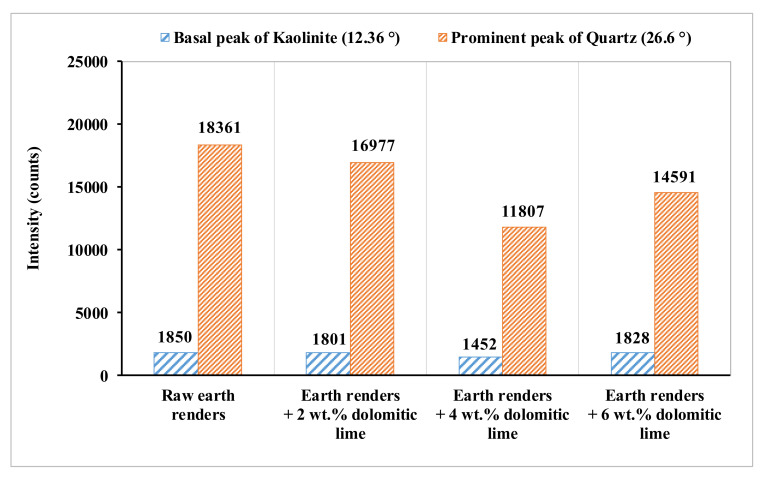
Evolution of the kaolinite basal and prominent quartz peaks versus dolomitic lime.

**Figure 8 materials-15-04014-f008:**
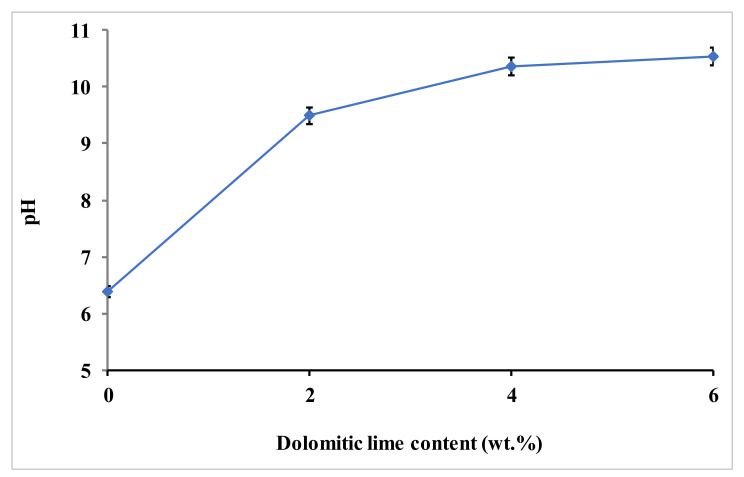
Variation of pH versus dolomitic lime.

**Figure 9 materials-15-04014-f009:**
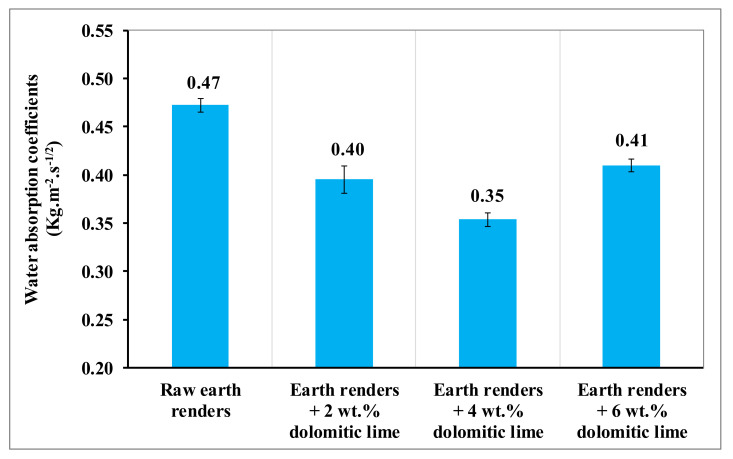
Coefficient of water absorption by capillarity of earth renders samples.

**Figure 10 materials-15-04014-f010:**
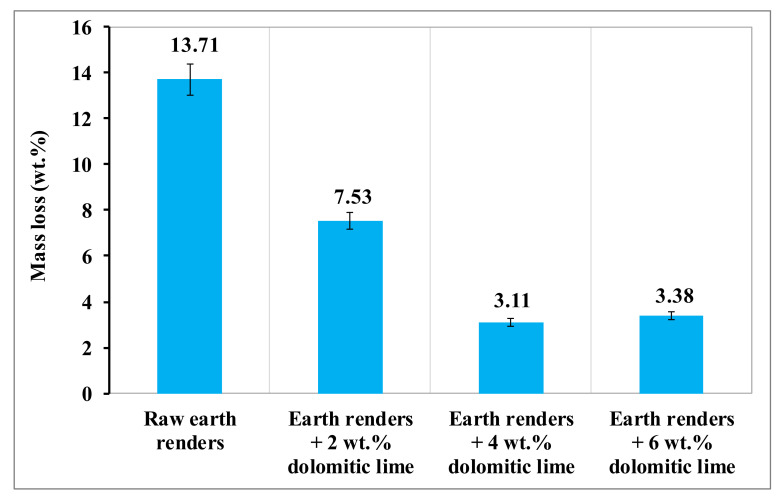
Mass loss of earth renders with dolomitic lime addition.

**Figure 11 materials-15-04014-f011:**
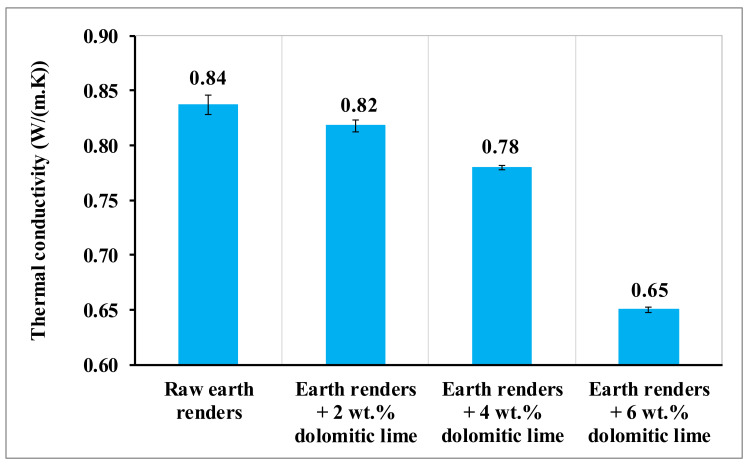
Thermal conductivity of earth renders with dolomitic lime additions.

**Figure 12 materials-15-04014-f012:**
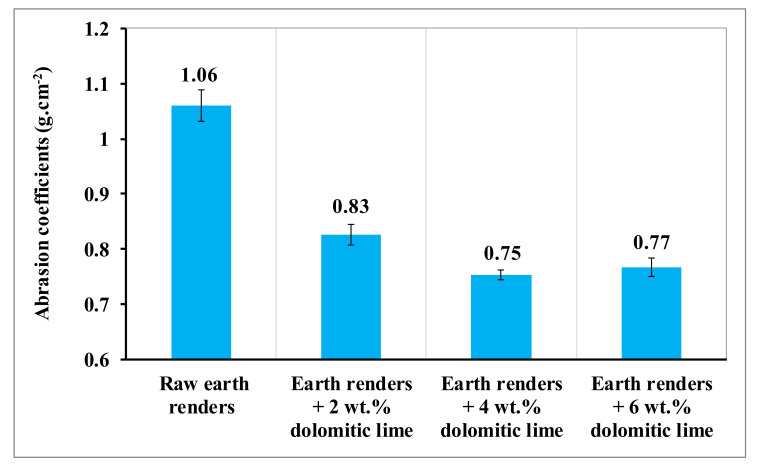
Abrasion coefficient of earth renders versus dolomitic lime.

**Figure 13 materials-15-04014-f013:**
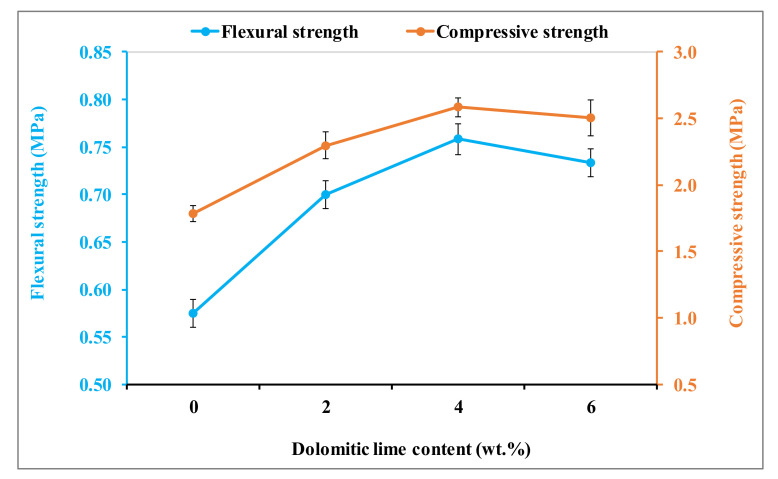
Flexural and compressive strengths of earth renders samples amended with dolomitic lime.

**Table 1 materials-15-04014-t001:** Geotechnical and mineralogical composition of raw soil [1].

Particles Size Distribution				Mineralogical Composition			Atterberg Limits			Methylene Blue Value	Clay Category
Clay (wt.%)	Silt (wt.%)	Fine sand (wt.%)	Coarse sand (wt.%)	Kaolinite (wt.%)	Quartz (wt.%)	Goethite (wt.%)	W_L_ (%)	W_P_ (%)	PI (%)	1.43 g/100 g	Sandy-loam soils and sensitive to water
40	10	34	16	62	31	2	44	21	23		

**Table 2 materials-15-04014-t002:** Proportions of earth rendering mortars.

Code	Earth Rendering Mortars Description	Clay (g)	Dolomitic Lime (g)	Water (mL)
RM	Raw earth rendering mortar	1500	0	495
M-DL_2_	Earth rendering mortar + 2 wt.% dolomitic lime	1470	30	485.1
M-DL_4_	Earth rendering mortar + 4 wt.% dolomitic lime	1440	60	475.2
M-DL_6_	Earth rendering mortar + 6 wt.% dolomitic lime	1410	90	465.3

**Table 3 materials-15-04014-t003:** Physical properties of earth renders amended with dolomitic lime.

Earth Rendering Mortars	Linear Shrinkage (%)	Apparent Density (g/cm^3^)	Accessible Porosity (%)
Raw earth renders	3.13 ± 0.03	1.72 ±0.01	3.57 ± 0.04
Earth renders + 2wt.% dolomitic lime	2.71 ± 0.02	1.69 ± 0.02	35.41 ± 0.01
Earth renders + 4wt.% dolomitic lime	1.88 ± 0.01	1.65 ± 0.03	36.12 ± 0.01
Earth renders + 6wt.% dolomitic lime	1.67 ± 0.02	1.63 ± 0.01	37.53 ± 0.02

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
