# Peer review of "Physical, Hydric, Thermal and Mechanical Properties of Earth Renders Amended with Dolomitic Lime"

_materials, 2022, doi:10.3390/ma15114014_

Round 1

Reviewer 1 Report

The first problem with this article is that much of it has already been published elsewhere and I think that part can therefore be quoted without being fully copied.

I think the authors must remove:

Fig.1, Table 2, Fig.2, Fig.3, Fig. 4, so that the whole section 2.1 can be rearranged as a short summary citing the previous article. Moreover, many sentences lack references.

The section 2.2. Manufacture of earth rendering mortars should be improved by specifying how many samples were prepared and analyzed with the different additions of Mg-lime.

The 2.3.1. Chemical, mineralogical and microstructural characterization of raw materials and earth renders, should be improved and the details of the analytical procedure further specified.

The results section starts with the characterization of dolomitic lime, but it is not clear whether this was done before, during or after carbonation. The presence of portlandite together with calcium oxide and Ca and Mg carbonates are strongly dependent on this aspect.

In addition, the anomalous formation of Ca-Mg hydrated silicates (pozzolanic reaction) only by mixing dolomitic lime and clay is not demonstrated:

  • The endothermic peak between 100-200 °C in Fig. 9 can be attributed to the clay.
  • The increase in pH is normal when increasing the amount of Ca and Mg hydroxides.
  • It is very unlikely that quartz, even if fine-grained, could be responsible for the pozzolanic reaction due to its high crystallinity.
  • In fig.10 the error bars are missing and we do not know how many samples were analyzed and the standard deviation. Here the authors stated “It is also observed that the intensity of the main quartz peak decreases up to 36 % with 4 wt.% addition of dolomitic lime while the basal kaolinite peak decreases only by 22 % for the same formulation.” But the characterization of the sample composition was done only after the setting reaction, so we have no idea of the previous percentage of quartz and kaolinite.
  • - The improvement in physical-mechanical properties can be explained simply by the formation of Ca-Mg carbonates.

Other minor and major inaccuracies are scattered throughout the text, but before addressing them I think the article needs a thorough revision and it is not suitable for publication.

Author Response

Comments and Suggestions for Authors

First of all, we would like to thank the reviewer for his review of the paper which will improve the scientific quality of the paper. We have taken into account all his comments. The new modifications appear into the revised manuscript in red colour.

The research design has been improved in order to make it appropriate. This could be seen in revised manuscript.

 We think that the conclusions are supported by the results.

In the revised manuscript the methods are now adequately described and the results are cleary presented.

The first problem with this article is that much of it has already been published elsewhere and I think that part can therefore be quoted without being fully copied.

This suggestion has been taken into account in the revised manuscript.

I think the authors must remove:

Fig.1, Table 2, Fig.2, Fig.3, Fig. 4, so that the whole section 2.1 can be rearranged as a short summary citing the previous article. Moreover, many sentences lack references.

This suggestion has been taken into account in the revised manuscript.

The section 2.2. Manufacture of earth rendering mortars should be improved by specifying how many samples were prepared and analyzed with the different additions of Mg-lime.

This suggestion has been taken into account in the revised manuscript.

The 2.3.1. Chemical, mineralogical and microstructural characterization of raw materials and earth renders, should be improved and the details of the analytical procedure further specified.

This suggestion has been taken into account in the revised manuscript.

The results section starts with the characterization of dolomitic lime, but it is not clear whether this was done before, during or after carbonation. The presence of portlandite together with calcium oxide and Ca and Mg carbonates are strongly dependent on this aspect.

The characterization of dolomitic lime was done after carbonation because the dolomitic lime used in this study was poorly hydrated and carbonated by air humidity.

In addition, the anomalous formation of Ca-Mg hydrated silicates (pozzolanic reaction) only by mixing dolomitic lime and clay is not demonstrated:

This has been taken into account in the manuscript. This is justified by scientific works published by Bernar et al :2017. (Bernard, E., Lothenbach, B., Rentsch, D., Pochard, I. Formation of magnesium silicate hydrates (M-S-H). Phys Chem Earth. 2017, 99, 142-157.) In addition, MSH could be formed due to the high MgO content (37.25 wt.%)

  • The endothermic peak between 100-200 °C in Fig. 9 can be attributed to the clay.

We are sorry. This has been taken into account in the revised manuscript.

  • The increase in pH is normal when increasing the amount of Ca and Mg hydroxides.

It’s true.

  • It is very unlikely that quartz, even if fine-grained, could be responsible for the pozzolanic reaction due to its high crystallinity.

We think that the reaction of fine quartz with portlandite and magnesium hydroxide could be responsible for the pozzolanic reaction because grinding the sample makes the surface of fine grains of quartz more active and promotes this reaction (Dao et al.2018) [24]. A soil that has a significant amount of fine quartz is conducive to pozzolanic reaction into portlandite and magnesium hydroxide.

  • In fig.10 the error bars are missing and we do not know how many samples were analyzed and the standard deviation. Here the authors stated “It is also observed that the intensity of the main quartz peak decreases up to 36 % with 4 wt.% addition of dolomitic lime while the basal kaolinite peak decreases only by 22 % for the same formulation.” But the characterization of the sample composition was done only after the setting reaction, so we have no idea of the previous percentage of quartz and kaolinite.

Mineralogical composition of sample is given in the manuscript: 31wt.% of quartz, 62wt.% of kaolinite and 2wt.% of goethite.

Reviewer’s suggestions have been taken into account in revised manuscript.

  • - The improvement in physical-mechanical properties can be explained simply by the formation of Ca-Mg carbonates.

It is possible that the improvement in physical-mechanical properties of earth renders could be explained by the formation of Ca-Mg carbonates but the contribution of CSH or MSH is very important because of there are cementitious mineral phases. It is observed that excessive amount of Ca-Mg carbonates could decrease these physical and mechanical properties of building materials.

Reviewer 2 Report

  1. Page 9, Does the linear shrinkage test follow any standard? If yes, please mark it clearly.
  2. Chapter 2.1 and Chapter 3.1 are discussions on the analysis results of raw materials, which should be grouped to facilitate readers’ comparison and make the article's content more concise.
  3. Page 14 mentions a value change between 100-200oC, which belongs to the water loss of CSH and MSH. In fact, the hydrate of CSH and MSH should belong to crystal water, and the weight change should be between 400-500oC. Please confirm it.
  4. Most of the experimental results in the article are explained by the CSH and MSH reactions of the materials. CSH and MSH should be able to be analyzed by XRD. Therefore, the samples should be subjected to XRD analysis.
  5. Page 18, “The apparent density of the earth renders samples decreases with dolomitic lime additions. This decrease is due to the replacement of the clayey earth particles by dolomitic lime powder less dense.” This description is not suitable. The specific density of dolomitic lime is around 3.2-3.4 and clay(kaolin) is around 2.65. So, the more dolomitic lime is added, the heavier the specific gravity should be.

Author Response

Comments and Suggestions for Authors

First of all, we would like to thank the reviewer for his review of the paper which will improve the scientific quality of the paper. We have taken into account all his comments. The new modifications appear into the revised manuscript in red colour.

The research design has been improved in order to make it appropriate. This could be seen in revised manuscript.

 We think that the conclusions are supported by the results because of the results are clearly presented according to the reviewer.

  1. Page 9, Does the linear shrinkage test follow any standard? If yes, please mark it clearly.

Linear shrinkage test is realized using a caliper according to the German standard (DIN 18947 (Deutsches Institut für Normung E.V.), Earth Plasters-Terms and definitions,
requirements, test methods. 2013). This has been taken into account in the revised manuscript.

  1. Chapter 2.1 and Chapter 3.1 are discussions on the analysis results of raw materials, which should be grouped to facilitate readers’ comparison and make the article's content more concise.

This has been taken into account in the revised manuscript.

  1. Page 14 mentions a value change between 100-200oC, which belongs to the water loss of CSH and MSH. In fact, the hydrate of CSH and MSH should belong to crystal water, and the weight change should be between 400-500oC. Please confirm it.

According to the litterature, crystal water lost for CSH and MSH  is around  150-300°C.

The references used to justify this are :

Rojas, M.F; Cabrera, J. The effect of temperature on the hydration rate and stability of hydration phases of metakaolin-lime-water systems. Cem. Concr. Resear. 2002, 32, 133 138.

Assal, H.H. Influence of silica fume on the properties of fired then water cured lime-rich
clay bricks. Sil Indus. 2003, 68(5-6), 55-60.

Bernard, E., Lothenbach, B., Rentsch, D., Pochard, I. Formation of magnesium silicate hydrates (M-S-H). Phys Chem Earth. 2017, 99, 142-157.

This has been taken into account in the revised manuscript.

  1. Most of the experimental results in the article are explained by the CSH and MSH reactions of the materials. CSH and MSH should be able to be analyzed by XRD. Therefore, the samples should be subjected to XRD analysis. CSH and MSH are amorphous minerals therefore they can not be identied by XRD but by IR or DSC/TGA.

CSH presents diffuse peaks with feeble intensity around 48°<2θ<50°. This has been corrected in revised manuscript.

  1. Page 18, “The apparent density of the earth renders samples decreases with dolomitic lime additions. This decrease is due to the replacement of the clayey earth particles by dolomitic lime powder less dense.” This description is not suitable. The specific density of dolomitic lime is around 3.2-3.4 and clay(kaolin) is around 2.65. So, the more dolomitic lime is added, the heavier the specific gravity should be.

We are very sorry. The decrease of apparent density is due to the increasing of accessible porosity. This has been corrected in revised manuscript.

Reviewer 3 Report

  1. It is recommended to describe the relevance of the ongoing research and explain the feasibility of using KOD clay in this article.
  2.   Please explain what is the purpose of this article for TGA studies for KOD clay and dolomitic lime?
  3.  Section 3.1, which contains a description of the characteristics of dolomitic lime, would logically be moved to section 2.3, which provides similar information for KOD clay
  4. By analogy with Figure 8, given for dolomitic lime, it is recommended to add similar information for KOD clay.
  5.  In section 3, it is recommended to provide a Table containing the component compositions of the experimental mixtures of the renders.

Author Response

Comments and Suggestions for Authors

First of all, we would like to thank the reviewer for his review of the paper which will improve the scientific quality of the paper. We have taken into account all his comments. The new modifications appear into the revised manuscript in red colour.

The introduction section has been rewritten taking into account reviewer’s suggestions and comments. The methods are also adequately described.

  1. It is recommended to describe the relevance of the ongoing research and explain the feasibility of using KOD clay in this article.

This recommendation has been taken into account in the revised manuscript.

  1.   Please explain what is the purpose of this article for TGA studies for KOD clay and dolomitic lime?

This analysis allows to record the mass variations during a thermal cycle. These variations are related to chemical reactions or to the departure of volatile constituents adsorbed or combined in a material and it also allows to study the behavior of some minerals according to the temperature. It is a complementary analysis allowing to identify the minerals.

  1.  Section 3.1, which contains a description of the characteristics of dolomitic lime, would logically be moved to section 2.3, which provides similar information for KOD clay

This suggestion has been taken into account in the revised manuscript.

  1. By analogy with Figure 8, given for dolomitic lime, it is recommended to add similar information for KOD clay.

Sorry for not putting the SEM-EDS images of the KOD clay in the manuscript because it has not been realized. This figure does not give important informations on raw material better than those of mineralogical characterization by XRD, IR and DSC/TGA.

  1.  In section 3, it is recommended to provide a Table containing the component compositions of the experimental mixtures of the renders.

This has been taken into account in the revised manuscript.
